# Protective MVA-ST Vaccination Robustly Activates T Cells and Antibodies in an Aged-Hamster Model for COVID-19

**DOI:** 10.3390/vaccines12010052

**Published:** 2024-01-03

**Authors:** Sabrina Clever, Lisa-Marie Schünemann, Federico Armando, Christian Meyer zu Natrup, Tamara Tuchel, Alina Tscherne, Malgorzata Ciurkiewicz, Wolfgang Baumgärtner, Gerd Sutter, Asisa Volz

**Affiliations:** 1Institute of Virology, University of Veterinary Medicine Hannover, 30559 Hanover, Germany; sabrina.clever@tiho-hannover.de (S.C.); lisa-marie.schuenemann@tiho-hannover.de (L.-M.S.); christian.meyer.zu.natrup@tiho-hannover.de (C.M.z.N.);; 2Department of Pathology, University of Veterinary Medicine Hannover, 30559 Hanover, Germanywolfgang.baumgaertner@tiho-hannover.de (W.B.); 3Pathology Unit, Department of Veterinary Science, University of Parma, 43121 Parma, Italy; 4Division of Virology, Department of Veterinary Sciences, LMU Munich, 80539 Munich, Germany; alina.tscherne@viro.vetmed.uni-muenchen.de (A.T.); gerd.sutter@lmu.de (G.S.)

**Keywords:** poxvirus, recombinant vaccine, preclinical model, vaccination elderly

## Abstract

Aging is associated with a decline in immune system functionality. So-called immunosenescence may impair the successful vaccination of elderly people. Thus, improved vaccination strategies also suitable for an aged immune system are required. Modified Vaccinia virus Ankara (MVA) is a highly attenuated and replication-deficient vaccinia virus that has been established as a multipurpose viral vector for vaccine development against various infections. We characterized a recombinant MVA expressing a prefusion-stabilized version of SARS-CoV-2 S protein (MVA-ST) in an aged-hamster model for COVID-19. Intramuscular MVA-ST immunization resulted in protection from disease and severe lung pathology. Importantly, this protection was correlated with a potent activation of SARS-CoV-2 specific T-cells and neutralizing antibodies. Our results suggest that MVA vector vaccines merit further evaluation in preclinical models to contribute to future clinical development as candidate vaccines in elderly people to overcome the limitations of age-dependent immunosenescence.

## 1. Introduction

Infection with SARS-CoV-2 in elderly people has been demonstrated to result in severe morbidity and mortality [1]. This was most obvious during the beginning of the SARS-CoV-2 pandemic in 2020, when the fatality rates were 8-fold higher in people older than 65 years of age [2]. In general, the clinical presentation of COVID-19 in elderly people is mainly associated with more severe grades of respiratory symptoms, such as cough and dyspnea, as well as more pronounced outcomes from other symptoms, including headache, fever, fatigue, anosmia and gastrointestinal symptoms [3]. 

In addition, the efficacy of COVID-19 vaccines against symptomatic infection was lower in older adults (aged ≥ 60 years) compared with younger individuals [4], confirming previous data from the evaluation of flu vaccines in the elderly [5]. While seroconversion has been confirmed in elderly humans after immunization with COVID-19 and flu vaccines, the mean geometric titer (GMT) of virus-specific antibodies is significantly lower compared to younger and young adult individuals. Reduced GMT levels have been associated with higher infection rates and more severe clinical disease outcomes in older adults [6,7,8]. However, clinical studies evaluating on the safety, immunogenicity and efficacy of COVID-19 vaccines are still very limited. 

The Syrian hamster model has been established as a surrogate model for COVID-19 in humans. Previous studies confirmed that SARS-CoV-2 infection in aged hamsters (28–40 weeks old), as opposed to 6–8-week-old adults, resulted in more severe disease outcomes, mimicking the disease phenotype seen in elderly humans [9,10,11].

Modified Vaccinia virus Ankara (MVA), a safety-tested and replication-deficient vaccinia virus, is licensed as a third-generation smallpox and Mpox vaccine and has a well-established record in clinical safety. The well-established safety profile of MVA has also been confirmed in immunocompromised persons and elderly individuals [12,13]. In addition, MVA also serves as an advanced vaccine technology platform for developing viral vector vaccines against infectious diseases and cancer (for a review, see [14]). Moreover, MVA is part of a licensed heterologous prime-boost vector vaccine against Ebola virus (EBOV), in combination with a recombinant adenovirus vector [15,16,17].

In recent studies, MVA has also been established as viral vector vaccine against SARS-CoV-2. In this context, all MVA-based COVID-19 candidate vaccines used the SARS-CoV-2 spike protein in a prefusion-stabilized conformation (ST-protein) as a vaccine antigen. These different MVA-COVID-19 vaccines confirmed their protective efficacy in different preclinical models [18,19,20,21]. 

Previously, we confirmed that an MVA-based candidate vaccine against COVID-19 expressing the prefusion-stabilized SARS-CoV-2 spike protein (MVA-ST) was immunogenic and effective when tested on k18-hACE2 mice (6–8 weeks old) and hamsters (10 weeks old) [21]. These results demonstrated safety, immunogenicity and protective outcomes in adult animals. Moreover, the safety and robust activation of the specific immune responses of this MVA-ST candidate vaccine have been confirmed in phase I clinical testing in humans (unpublished data). 

Based on these previous data, we hypothesize that our MVA-ST candidate vaccine might also be suitable as a safe and effective COVID-19 vaccine in an aged-hamster model for SARS-CoV-2.

Here, we set out to evaluate the safety, immunogenicity and efficacy of MVA-ST vaccination in aged hamsters as a proof of concept. Prime-boost intramuscular MVA-ST immunization sufficiently prevented severe disease outcomes in aged hamsters. This protective capacity was correlated with the robust activation of neutralizing antibodies and T cells in MVA-ST-vaccinated aged hamsters.

## 2. Materials and Methods

Ethics statement. Male Syrian hamsters (66 weeks old, Mesocricetus auratus; breed RjHan:AURA) were bought from the Janvier Labs (SAINT BERTHEVIN CEDEX, France). We housed the hamsters under specified pathogen-free conditions, with free access to water and food. Before the immunization with non-recombinant MVA vaccine or MVA-ST vaccine, the hamsters adapted to our stables for at least one week. Our hamster studies fully meet the requirements of the European and national regulations for animal experimentation (European Directive 2010/63/EU; Animal Welfare Acts in Germany) and Animal Welfare Act, approved by the Niedersächsisches Landesamt für Verbraucherschutz und Lebensmittelsicherheit (LAVES) Lower Saxony, Germany). After SARS-CoV-2 challenge, the animals were housed in individually ventilated cages (IVCs; Tecniplast, Buguggiate, Italy) in approved BSL-3 facilities. All animal and laboratory work involving infectious SARS-CoV-2 was conducted and facilitated in a biosafety level (BSL)-3e laboratory and facilities at the Research Center for Emerging Infections and Zoonoses, University of Veterinary Medicine, Hannover.

Immunization experiments in hamsters. Male Syrian hamsters (66 weeks old, Mesocricetus auratus; breed RjHan:AURA) were immunized with 10^8^ PFU recombinant MVA-ST or empty-MVA-vector control intramuscularly into the quadriceps muscle of the left hind leg. Second immunization (Boost immunization) was applicated 21 days later. We closely observed and monitored the hamsters after the immunizations for well-being, health constitution and clinical signs, represented with a clinical score. We also monitored the body weights daily. For further analysis, we collected blood at different time points (days 0, 21, 42 and 55) after the immunization. Serum was prepared by centrifugation of the coagulated blood at 1300× *g* for 5 min in serum tubes (Sarstedt AG & Co., Nümbrecht, Germany). Serum samples were then stored at −80 °C until further evaluations.

SARS-CoV-2-infection experiments in Syrian hamsters. We performed challenge infection by the intranasal route, with 1 × 10^4^ TCID_50_ of SARS-CoV-2 (Isolate Germany/BavPat1/2020, NR-52370) received from BEI Resources, NIAID, NIH, under anesthesia. After respiratory challenge, the hamsters were monitored at least twice per day for well-being, health constitution and clinical signs using a clinical score sheet by allocating them to one of the following categories of COVID-19-disease-specific symptoms: Cardiovascular system, fur/skin condition, lower respiratory tract, upper respiratory tract, environment social behavior/general condition/locomotion and neurological scoring. Body weights were checked daily. 

Virus. The SARS-CoV-2 (Isolate Germany/BavPat1/2020) received from BEI Resources, NIAID, NIH was amplified in VeroE6 cells (ATCC #CRL-1586) in DMEM (Sigma-Aldrich GmbH, St Louis, MO, USA), including 2% fetal bovine serum, 1% penicillin–streptomycin and 1% L-glutamine at 37 °C. Experiments including SARS-CoV-2-infection were conducted in biosafety-level-3 laboratories at the RIZ, University of Veterinary Medicine Hannover, Germany. The recombinant MVA candidate vaccine expressing a prefusion-stabilized version of SARS-CoV-2 spike protein (MVA-ST) was generated as previously described [21]. The MVA-ST was amplified on DF-1 cells and purified by sucrose-gradient as previously described [21]. 

Plaque-reduction-neutralization test (PRNT_50_). Plaque-reduction-neutralization tests (PRNT_50_) were performed to evaluate the titers of neutralizing antibodies against SARS-CoV-2 in serum samples. The SARS-CoV-2 (Isolate Germany/BavPat1/2020) obtained from BEI Resources, NIAID, NIH was used for the infection of VeroE6 cells in the PRNT_50_ assay. Heat-inactivated serum samples were plated as duplicates in 2-fold dilutions in 50 µL DMEM on 96-well plates. In total, 50 μL of SARS-CoV-2 (600 TCID_50_) was added per well and incubated at 37 °C for 1 h. After incubation, the mixture was placed on VeroE6 cells using 96-well plates and incubated for 45 min. A total of 100 µL DMEM mixed 1:1 with Avicel RC-591 (Dupont, Nutrition & Biosciences, Brabrand, Denmark) was layered on each well and the plates were incubated for 24 h at 37 °C. Finally, the cells were fixed with 4% formaldehyde/PBS for further staining. To this end, a polyclonal rabbit antibody targeting the SARS-CoV-2 nucleoprotein (clone 40588-T62, Sino Biological, Wayne, PA, USA) was used. A secondary peroxidase-labeled goat anti-rabbit IgG (Dako, Agilent) was used to develop a signal after adding a precipitate forming TMB substrate (True Blue, KPL SeraCare, Milford, MA, USA). Counts of cells infected with SARS-CoV-2 were measured with the ImmunoSpot^®^ reader (CTL Europe GmbH, Bonn, Germany). Using the BioSpot™ Software Suite, the serum-neutralization titer (PRNT_50_) was calculated. To this end, the reciprocal of the highest serum dilution leading to a reduction of >50% in the plaque formation by SARS-CoV-2 infection was used.

Measurement of viral burden. To analyze the viral load in nasal or oropharyngeal swabs, swabs were taken either from the nose or from the oropharynx by rotating the tips of the swabs. After sampling, swabs were then dissociated and lysed in 1 mL DMEM containing P/S (penicillin and streptomycin, Gibco). To analyze the viral load in the lungs of infected hamsters, tissue samples were taken after euthanasia and further prepared by homogenization in 1 mL DMEM containing P/S (penicillin and streptomycin, Gibco). Homogenization was performed with the TissueLyser-II (Qiagen, Hilden, Germany) and the homogenate was stored at −80 °C until further use. To determine the titers of infectious SARS-CoV-2, lung homogenates or swab lysates, in DMEM containing 5% FBS, were incubated in serial 10-fold dilutions on Vero cells in 96-well plates. After four days of incubation at 37 °C, the median tissue culture infectious dose (TCID_50_ units/mL) was calculated using the Reed–Muench method based on cytopathic effects in the cells. For statistical analysis, the data points of samples that did not induce cytopathic effects were changed to half of the detection limit. To determine levels of viral RNA of SARS-CoV-2 in the lungs, RT-qPCR (quantitative real-time reverse transcription PCR) analysis targeting different genes of SARS-CoV-2 were conducted. Using the KingFisher Flex, RNA was extracted from lung tissue samples with the NucleoMag RNA kit according to the manufacturer’s protocol. For SARS-CoV-2 RNA amplification, the Luna^®^ Universal Probe One-Step RT-qPCR Kit (NEB #E3006, New England Biolabs GmbH, Frankfurt am Main, Germany) was used in a CFX96-Touch Real-Time PCR system (Bio-Rad, Feldkirchen, Germany). The RT-qPCR assay specific to the RdRp gene of SARS-CoV-2 and recommended by the WHO was used: SARS-2-IP4, forward primer (5′-GGT AAC TGG TAT GAT TTC G-3′), reverse primer (5′-CTG GTC AAG GTT AAT ATA GG-3′) and probe (5′-TCA TAC AAA CCA CGC CAG G-3′ [5′]FAM [3′]BHQ-1)]. The RT-qPCR assay specific to the subgenomic E of SARS-CoV-2 used the following primers: forward primer (5′-ATATTGCAGCAGTACGCACACA-3′), reverse primer (5′-CGATCTCTTGTAGATCTGTTCTC-3′) and probe (5′-ACACTAGCCATCCTTACTGCGCTTCG-3′ [5′]FAM [3′]BBQ]. The RT-qPCR assay specific to the E gene of SARS-CoV-2 used the following primers: forward primer (5′-ACAGGTACGTTAATAGTTAATAGCGT-3′), reverse primer (5′-ATATTGCAGCAGTACGCACACA-3′) and probe (5′-ACACTAGCCATCCTTACTGCGCTTCG-3′ [5′]FAM [3′]BBQ]. The PCR program included reverse transcription at 50 °C for 10 min, denaturation at 95 °C for 1 min and 44 cycles of 95 °C for 10 s (denaturation) and 56 °C for 30 s (annealing and elongation). The relative fluorescence units (RFU) were measured at the end of the elongation step. The sample Ct value was correlated to a standard RNA transcript and the quantity of viral copy numbers per µL of total RNA was calculated.

Histological evaluation of lung pathology in hamsters. Injection and plunging of 10% buffered formalin in the lung were performed for fixation purposes. Left lung lobe tissue samples were further embedded in paraffin and 2–3-micron-thick sections were generated. Hematoxylin and eosin (HE) staining was performed for the evaluation of lesions in the lung. Analysis was conducted blinded with a semi-quantitative scoring system. Briefly, the evaluation included assessment of alveolar lesions (inflammation, regeneration, necrosis/desquamation and loss of alveolar cells, atypical large/syncytial cells, intraalveolar fibrin, alveolar edema, hemorrhage), airway lesions (inflammation, necrosis, hyperplasia) and vascular lesions (vasculitis, perivascular cuffing, edema, and hemorrhage). The total scores reflect the sum of all scores in the respective lung-anatomical compartments. Details on the scoring system were described previously [22].

Immunohistochemistry targeting the Nucleocapsid of SARS-CoV-2. Formalin-fixed, paraffin-embedded lung tissue samples were stained using a monoclonal mouse antibody (Sino Biological, 40143-MM0) against the nucleoprotein of SARS-CoV-2 and the Dako EnVision+ polymer system (Dako Agilent Pathology Solutions) as described previously [23]. 

To quantify SARS-CoV-2 NP immunolabeled cells in pulmonary tissues, slides were digitized using a slide scanner (Olympus VS200 Digital; Olympus Deutschland GmbH, Hamburg, Germany). QuPath (version 0.3.1) was used to perform image analysis [24]. Detection of lung tissue was performed automatically using digital thresholding. Images of whole slides of the entire left lung were evaluated. Blood vessels as well as artifacts were subtracted from the total lung tissue as they were indicated as ROIs. Automated positive cell detection was used, as previously described [25], to quantify the immune-stained cells in the lung tissues. This was based on marker-specific thresholding parameters.

Enzyme-linked Immunospot (ELISpot). Hamsters were euthanized on day 6 after challenge infection and splenocytes were isolated immediately. To this end, spleens were smashed on a 70 µm strainer (Falcon^®^, Sigma-Aldrich, Taufkirchen, Germany) and flushed and solved with RPMI-10 (RPMI 1640 medium containing 10% FBS, 1% penicillin–treptomycin, 1% HEPES; Sigma-Aldrich, Taufkirchen, Germany). Lyses of red blood cells were conducted with Red Blood Cell Lysis Buffer (Sigma-Aldrich, Taufkirchen, Germany). After one washing step, cells were resuspended in RPMI-10. In total, 2 × 10^5^ splenocytes were counted with the MACSQuant (Miltenyi Biotec B.V. & Co., KG, Bergisch Gladbach, Germany) and seeded per well in 96-well round-bottom plates (Sarstedt, Nümbrecht, Germany). For stimulation, three different peptide pools of overlapping peptides obtained by JPT Peptide Technologies (Berlin, Germany) were used. Each peptide consisted of 15 amino acids (15 mers) overlapping in 11 amino acids. Two of the peptide pools consisted of 157 and 158 overlapping peptides (1 µg peptide/mL RPMI 1640) comprising the whole spike glycoprotein of SARS-CoV-2. One additional peptide pool was used for stimulation, which consisted of 59 overlapping peptides (1 µg peptide/mL RPMI 1640) comprising the whole nucleocapsid protein of SARS-CoV-2 (BEI Resources, NIAID, NIH). This antigen is not present in the evaluated vaccine. Positive and negative controls were created by stimulation of the cells with phorbol myristate acetate and ionomycin (PMA, SIGMA-ALDRICH, Taufkirchen, Germany), as well as the use of non-stimulated cells. Plates with PVDF membranes (Mabtech, Nacka, Sweden) were coated with a hamster-specific anti- IFN-γ monoclonal antibody (Mabtech, Nacka, Sweden). Cells were then placed on the coated plates and incubated for 36 h at 37 °C. The inoculum was removed and biotinylated anti-IFN-γ monoclonal antibody was added. After incubation, streptavidin ALP was added, followed by BCIP/NBT-plus substrate. Washing steps were also performed between all steps. The generated spots were scanned and counted with the automated ELISpot Reader ImmunoSpot S6 ULTIMATE UV Image Analyzer (Immunospot, Bonn, Germany) and further analyzed with ImmunoSpot 7.0.20.1 software.

Flow cytometry. For phenotype characterization, cells were isolated from lungs via digestion using RPMI 1640 supplemented with 10% FBS, 1 mg/mL collagenase (GENAXXON bioscience GmbH, Ulm) and 0,5 mg/mL DNase (Roche, Merck KGaA, Darmstadt) for 30 min at 37 °C. Subsequently, digested lungs were smashed and flushed with RPMI-10 (RPMI 1640 medium containing 10% FBS, 1% penicillin–streptomycin; Sigma-Aldrich, Taufkirchen, Germany) through a 70 µm strainer (Falcon^®^, Sigma-Aldrich, Taufkirchen, Germany). Incubation with Red Blood Cell Lysis Buffer (Sigma-Aldrich, Taufkirchen, Germany) lysed red blood cells. After a washing step, cells were resuspended in RPMI-10 medium. The cells were counted with the MACSQuant (Miltenyi Biotec B.V. & Co. KG, Bergisch Gladbach, Germany) and dead cells were visualized by staining 4 × 10^5^ lung cells with LIVE/DEAD fixable Near IR stain kit (InvitrogenTM, Thermo Fisher Scientific, Waltham, MA, USA) according to the manufacturer’s instructions, followed by fixation using 4% formaldehyde/PBS for 20 min. In total, 2 × 10^5^ cells were stained with antibodies against CXCR3 (RRID: AB_2743928), CD4 (RRID: AB_464894) and CD3 (RRID:AB_10841760) in a total volume of 200 µL for 30 min. After washing, cells were measured using the MACSQuant and quantified by fluorescence, measured by median fluorescence intensity (MFI) (Miltenyi Biotec B.V. & Co., KG, Bergisch Gladbach, Germany).

Analysis of IL-7 by Real-time-PCR. Extracted RNA from lung tissue samples was amplified in a CFX96-Touch Real-Time PCR system (Bio-Rad, Feldkirchen, Germany) using the commercially available Luna^®^ Universal One-Step RT-qPCR Kit (NEB #E3005, New England Biolabs GmbH, Frankfurt am Main, Germany). The RT-qPCR assay specific to the hamster IL-7 used the following primers: forward primer (5′-ATCCAAGCCACAAAAATAAAGCC-3′) and reverse primer (5′-TTTCTTGCTGTCACTGCTTTG-3′). To normalize values, ß-Actin was used as housekeeping gene using forward and reverse primers (5′-CCAAGGCCAACCGTGAAAAG-3′ and 5′-ATGGCTACGTACATGGCTGG-3′, respectively).

The PCR program included reverse transcription at 55 °C for 10 min, denaturation at 95 °C for 1 min and 44 cycles of 95 °C for 20 sec (denaturation) and 56 °C for 30 s (annealing and elongation). The relative fluorescence units (RFU) were measured at the end of the elongation step. Amount of IL-7 relative to the housekeeping gene was calculated using the ∆Ct method.

Statistical analysis. Data were prepared using GraphPad Prism 9.0.0 (GraphPad Software Inc., San Diego, CA, USA). Data sets were analyzed for normal distribution using D’Agostino-and-Person test or Shapiro–Wilk test. Normally distributed data were analyzed for differences using *t*-test or two-way ANOVA and expressed with mean. Non-normally distributed data were analyzed for differences using Mann–Whitney test and Friedman test and expressed as median. A *p* value < 0.05 indicates the threshold for statistical significance.

## 3. Results

### 3.1. MVA-ST Vaccination Induced Protection after SARS-CoV-2 Challenge in Aged Syrian Hamsters

The Syrian hamsters aged 66 weeks were vaccinated with 10^8^ PFU of empty MVA-vector control or MVA-SARS-2-ST (MVA-ST) via the intramuscular route. Twenty-one days later, all the hamsters were boosted (Figure 1A). The safety and immunogenicity were analyzed as established previously [21]. No clinical disease outcome or weight loss were detected in the MVA-control-vaccinated or in the MVA-ST-vaccinated aged hamsters when monitored over a time period of 57 days before SARS-CoV-2 challenge infection (Appendix A). No SARS-CoV-2-neutralizing antibodies were determined in the sera from the aged hamsters that received the empty MVA-vector control vaccine (<1:20, Figure 1B). In contrast, all the MVA-ST vaccinated animals produced easily detectable neutralizing antibodies with an average PRNT_50_ titer of 1:350 as early as 3 weeks after priming (Figure 1B). After the boost vaccination on day 21, we measured higher amounts of SARS-CoV-2-neutralizing antibodies in all the sera from the aged hamsters vaccinated with MVA-ST, with an average titer of 1:600 PRNT_50_ (Figure 1B). 

After intranasal SARS-CoV-2 infection on day 49 after the initial vaccination, all the hamsters lost about 5% of their body weight 2 days post-infection (dpi). The MVA-vaccinated control hamsters continued to show progressive weight loss until 6 dpi, losing approximately 15% of their original body weight, as measured on the day of the challenge. In contrast, no further loss of body weight was identified for the MVA-ST-vaccinated hamsters (Figure 1C). The control animals developed significant clinical symptoms with enhanced breathing, scruffy fur and reduced activity, culminating on day 6, with a total score of 6. Minimal COVID-19-specific symptoms were observed in the MVA-ST-vaccinated animals with cumulative scores ranging between 0.5 and 1 on days 4–6 (Figure 1D), mainly resulting in ruffled fur, without any respiratory symptoms. At 3 dpi, significant levels of infectious SARS-CoV-2 were detected in the oropharyngeal and nasal swabs of the control hamsters (median 6.3 × 10^3^ TCID_50_ in oropharyngeal, median 1.99 × 10^3^ in nasal swabs; Figure 2A,B). The MVA-ST-vaccinated animals had significantly reduced titers of infectious SARS-CoV-2 in their oropharyngeal swabs (median 2 × 10^2^ TCID_50_; Figure 2A) and nasal swabs (median 4.7 × 10^1^ TCID_50_; Figure 2B).

All the animals were euthanized at 6 dpi, and blood samples, lungs and spleens were taken for further analysis. In the lungs of the empty MVA-vector-vaccinated hamsters, we detected substantial titers of infectious SARS-CoV-2 (median 3.9 × 10^3^ TCID_50_), but no titers of SARS-CoV-2 were measured in the lungs of the MVA-ST-immunized hamsters (Figure 2C), as further confirmed by the qRT-PCR analysis (Figure 2D–F).

The pathohistological analysis of the hematoxylin-and-eosin-stained lung sections revealed large areas of lung consolidation and inflammation in the control hamsters (Figure 3A). In the alveolar lesions, we detected the accumulation of neutrophils and mononuclear cells, which also expanded to the alveolar septa and filled the alveolar lumina. These pathological changes were significantly reduced or nearly absent in the MVA-ST-vaccinated animals (Figure 3A,B). In the lungs of the MVA-vaccinated hamsters, we detected epithelial hyperplasia in the conductive airways and moderate numbers of macrophages, degenerated heterophils and sloughed epithelial cells within the airway lumen. Similar lesions in the conductive airways were significantly reduced or absent in the MVA-ST-vaccinated hamsters (Figure 3A,B). In the lungs of the MVA-control-vaccinated hamsters, we detected significantly marked vascular lesions compared to the MVA-ST vaccinated hamsters, in which they were almost absent. The vascular lesions mainly consisted of histiocytic–heterophilic perivascular and intramural infiltrates, including abundant perivascular edema and perivascular cuffing (Figure 3A,B). Most importantly, the alveoli of the MVA-vaccinated hamsters were completely obliterated by high numbers of heterophils, macrophages, extravasated erythrocytes and occasional fibrin or edema. On the other hand, the MVA-ST-vaccinated hamsters displayed normal alveoli, very often free of inflammatory changes, with a significantly lower histopathological score (Figure 3A,B). Notably, both groups occasionally showed alveolar septa covered by proliferating type II pneumocytes.

The immune staining specific to the nucleoprotein of SARS-CoV-2 showed high amounts of nucleoprotein-positive cells in the lungs of the control-vaccinated animals (MVA), in contrast to the significantly lower amounts of positive cells in the hamsters vaccinated with MVA-ST (Figure 3C).

### 3.2. MVA-ST Induced Immune Responses Correlated with Protection against SARS-CoV-2 Infection in Aged Syrian Hamsters

After the challenge infection, we detected levels of virus-neutralizing antibodies even in the sera of the control-vaccinated hamsters (mean 1:4200 PRNT_50_). In the MVA-ST vaccinated hamsters, the titers significantly increased to average PRNT_50_ titers of 1:8400 (Figure 4A). 

To evaluate the activation of theSARS-CoV-2-specific T cells, we prepared splenocytes and lung cells. The splenocytes were restimulated with pools of overlapping peptides comprising either the S1 or S2 subunit of the SARS-CoV-2 S-protein or the SARS-CoV-2 N-protein (Appendix A) and analyzed using ELISPOT assays. The MVA-ST vaccination activated robust numbers of S1-specific T cells, with a mean number of 1011 IFN-γ spot-forming cells (SFC) in the 10^6^ splenocytes (Figure 4B), whereas significantly lower numbers of these cells were detected in the control hamsters (mean 163 IFN-γ SFC; Figure 4B). The MVA-ST-vaccinated hamsters also displayed substantial levels of S2-specific T cells (mean 2048 IFN-γ SFC), while these cells were again significantly lower in the control animals (mean 1371 IFN-γ SFC). Comparable levels of T cells specific to the SARS-CoV-2-N were detected in both groups (MVA: mean 157 IFN-γ SFC and MVA-ST: mean 332 IFN-γ SFC; Figure 4B).

We used a panel of hamster reactive antibodies against specific markers (CD3, CD4, CXCR3 [26]) to characterize the T cells in the lungs by FACS analysis. The MVA-ST-vaccinated hamsters displayed significantly increased levels of T cells defined as CD3+ cells compared to the control animals (median 34% vs. 26%; Figure 5A). Similarly, significantly higher levels of CD4+ T cells were present in the MVA-ST-vaccinated animals than in the control animals (median 31% vs. 25%; Figure 5B). The CXCR3+ CD4+ T cells were also significantly increased in the MVA-ST-vaccinated hamsters (median 0,26%) compared to the control hamsters (median 0.035%; Figure 5C). As a marker for the reactivity of the immune system, we also measured increased levels of IL-7 in the lungs from the MVA-ST-vaccinated animals (median 4.45 × 10^4^; Figure 5E) compared with the control hamsters (median 2.62 × 10^6^). We then analyzed the expression of TCR, as measured by CD3 expression on the CD4+ T cells; the control hamsters demonstrated significantly higher expressions of TCR (5,8 median fluorescence intensity (MFI); Figure 5D) than the MVA-ST-vaccinated animals (median 4.3 MFI; Figure 5D). 

In summary, the MVA-ST vaccination protected the aged hamsters from COVID-19 and activated robust levels of SARS-CoV-2-specific immune responses. 

## 4. Discussion

Aging occurs alongside significant modifications in the immune system, leading to the reduced reactivity of immune responses and increased susceptibility to infectious diseases. In general, the outcome of such altered immune reactivity in an aged immune system is summarized under the term immunosenescence. Immunosenescence is also a known problem for successful vaccination in elderly people [27]. In this proof-of-concept study, we evaluated the safety, immunogenicity and efficacy of the well-established MVA-ST vaccine against COVID-19 in an aged-hamster model. To this end, we used 66-week-old hamsters, which correlates to about 65 years in human age [28]. In agreement with data from previous studies evaluating the effects of MVA-based COVID-19 vaccines in different preclinical models using different vaccination schedules [18,19,20,29], we confirmed the activation of SARS-CoV-2-specific immune response and protection in the aged MVA-ST-vaccinated hamsters compared to the control aged hamsters, which had been vaccinated with an empty MVA vector. Our results demonstrated the robust activation of SARS-CoV-2-specific antibodies and T cells and significantly reduced viral loads in the upper and lower respiratory tract in the MVA-ST-vaccinated aged hamsters. Despite the robust activation of SARS-CoV-2-specific immune responses in the MVA-ST-vaccinated aged hamsters following the infection challenge, we detected minimal disease outcomes with marginal body weight loss, clinical symptoms and viral shedding from the upper respiratory tract. This is in line with data from humans, in whom breakthrough infections with mild clinical disease outcomes have been confirmed in vaccinated people with a mean age of 71.1 [30,31]. The question of whether this minimal outcome of clinical disease in the MVA-ST-vaccinated aged hamsters might be the consequence of immunosenescence has to be addressed in future studies. Here, a study directly comparing the effects of MVA-ST vaccinations in aged and adult hamsters is essential to shed light on the potential contribution of the aged immune system. This is also true for the control hamsters. An effect of immunosenescence can also be hypothesized for the aged hamsters vaccinated with the empty MVA vector, since we observed significant SARS-CoV-2-disease outcomes, confirming previous studies using the aged-hamster model [9,10,11]. Again, whether the more obvious outcome of SARS-CoV-2 disease is a consequence of immunosenescence in aged hamsters needs to be evaluated in future studies, including direct side-by-side comparisons of different aged hamsters. The clinical disease outcome, as measured in the control-vaccinated hamsters and also, to a minimal extent, in the MVA-ST-vaccinated hamsters, can be correlated with SARS-CoV-2-intranasal-challenge infection, since we did not detect any clinical disease outcomes in the prior observation period. In our study, we also detected pronounced lung pathology, with massive extents of alveolar inflammation and damage in the control hamsters, confirming other studies characterizing SARS-CoV-2 pathogenesis in aged hamsters [9,10,11]. Again, the outcome of lung pathology is induced by SARS-CoV-2 infections, since other studies did not observe relevant pathology in the lungs of aged hamsters as background lesions. Similarly, in elderly people, SARS-CoV-2 infection often results in severe disease, with more pronounced lung pathology [32,33]. In addition to minimal clinical symptoms and reduced viral shedding from the upper respiratory tract early during infection, we detected no viral load and significantly reduced levels of lung damage in the aged MVA-ST-vaccinated hamsters compared to the control hamsters at the end of the experiment on day 6 post-infection, indicating a robust protective capacity. This is in line with previous studies, confirming the robust immunogenicity and efficacy of these MVA-based candidate vaccines against COVID-19. An interesting aspect of these studies is that MVA-ST vaccination also resulted in robust immunogenicity and protection against different SARS-CoV-2 viral variants [18,20,29]. Since this is also an extraordinarily important aspect of safe and effective COVID-19 vaccines for elderly individuals, it will also be important to evaluate the effects of MVA-ST vaccination in aged hamsters against different SARS-CoV-2 variants. 

The MVA is an attenuated, replication-deficient candidate vaccine with a well-established safety profile and excellent immunostimulatory effects. It is already confirmed as a smallpox vaccine, having been successfully administered to persons at risk, e.g., with atopic dermatitis or infected with HIV [12,34,35]. Thus, MVA seems highly suited to overcoming the limitations of vaccination in the elderly. This has already been successfully confirmed for an MVA-based vaccine against influenza in a clinical phase I study [13]. 

In line with clinical data from flu vaccine development, we confirmed the suitability of our MVA-ST vaccine in aged hamsters as a surrogate model for COVID-19 in humans. This was emphasized by the strong activation of neutralizing antibodies in the MVA-ST-vaccinated animals, even before the challenge infection. After the SARS-CoV-2 challenge, we also detected SARS-CoV-2-neutralizing antibodies in the control animals, despite the absence of obvious protective efficacy, possibly resulting from fulminant viral infection in the upper respiratory tract. Another hypothesis is that these increased titers of SARS-CoV-2-specific antibodies are involved in the outcome of severe COVID-19 disease. This has been demonstrated in elderly people with more severe disease who exhibited higher antibody reactivity [36]. In this context, cellular immunity was correlated with more robust and solid protection, independently of humoral immunity [37].

Notably, compared to the control hamsters, we demonstrated that MVA-ST-vaccinated elderly hamsters mounted a robust activation of T cells. Especially within the lung, the main target organ for COVID-19, we detected significantly increased titers of activated CD4+ T cells, as shown for CXCR3+ cells. Such strong immunostimulation is a well-known characteristic of MVA-based vaccines and is mainly correlated with the potent activation of innate immune signaling, leading to chemokine and cytokine induction, generating an optimal immunological milieu for antigen-specific immunity [38]. To support this, MVA-ST-vaccinated hamsters mounted significantly increased levels of Interleukin-7 (IL-7). The diminished activation of proinflammatory cytokines and, in particular, IL-7, have been confirmed as being involved in the outcome of immunosenescence [39]. Here, IL-7 is important and critically involved in the homeostasis of the immune system [40]. Reduced levels of IL-7 have been correlated with the decline and dysfunction of the immune system [41]. By contrast, the robust activation of IL-7 signaling positively affects the maturation and activation of adaptive immune responses, including B cells and T cells [42,43]. In line with this, the advanced activation of IL-7 might explain the improved activation of antigen-specific T cells in the MVA-ST-vaccinated animals. This was further confirmed not only by the increased levels of CD4+ T cells lacking TCR on the surface, indicating the activation of antigen-specific T cells [44], but also when we detected significantly increased levels of S1- and S2-specific T cells in the spleens of the MVA-ST-vaccinated hamsters. Future studies will be of interest to characterize the role and kinetics of these SARS-CoV-2 S-specific T cells in more detail concerning the outcomes of infection and/or vaccine induced protection.

In summary, our aged-hamster model confirmed the robust activation of SARS-CoV-2-specific immunity after MVA-ST-immunization, which is probably involved in protection against severe clinical disease outcomes. However, to directly correlate the outcomes of MVA-ST induced immunogenicity and efficacy with age-related changes in the immune system and the possible impact of immunosenescence in an aged-hamster model, follow-up studies including hamsters of different ages with and without SARS-CoV-2 infection need to be conducted.

## 5. Conclusions

In conclusion, our findings from this proof-of-concept study further indicate that the success of MVA-ST vaccination is not hampered in aged hamsters. Our results further support MVA-vector-vaccine-induced safety, immunogenicity and efficacy in elderly individuals as an important aspect of population-based immunity against COVID-19 and, probably, against other betacoronavirus-specific diseases, such as the Middle East Respiratory syndrome. We therefore highlight a promising approach for developing MVA-based vaccination for elderly people that results in the robust activation of antigen-specific T cells as a strategy to overcome immunosenescence, as well as other pathogens. 

## Figures and Tables

**Figure 1 vaccines-12-00052-f001:**
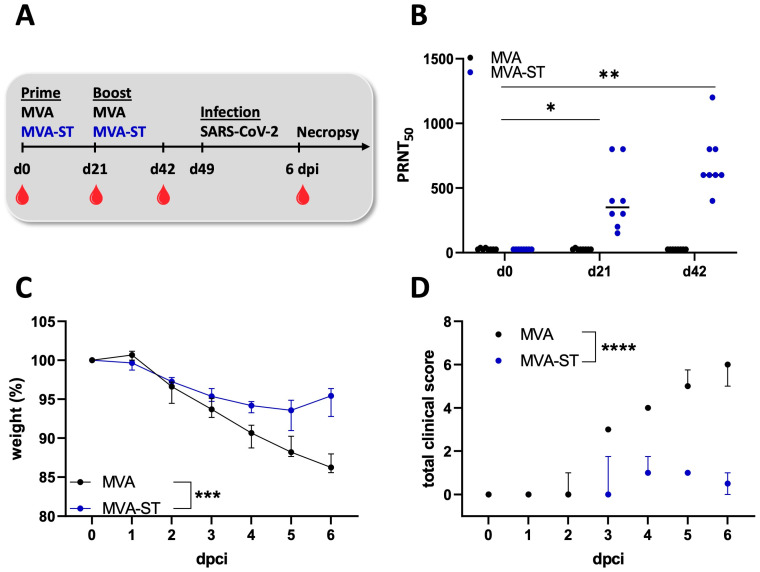
Immunogenicity and efficacy of MVA-ST immunization in aged hamsters. (**A**) Animals were vaccinated twice via the intramuscular route with 10^8^ PFU MVA-ST (*n* = 7) over a 21-day interval. Blood samples were taken at day 0, at day 21 after first immunization (prime) or at day 42 after first immunization (day 21 after second immunization) and 6 days post-challenge. Hamsters vaccinated with empty MVA vector (MVA, *n* = 7) served as controls. Body weight changes, clinical scores, viral loads, pathology and immunogenicity were determined. (**B**) Sera were analyzed for SARS-CoV-2-neutralizing antibodies using plaque-reduction assay (PRNT_50_) on days 0, 21 and 42 after initial vaccination. (**C**) After challenge infection on day 49, changes in body weights were monitored daily. (**D**) A clinical score sheet was used to monitor spontaneous behavior, clinical disease and general condition. Differences between groups were analyzed by Dunn’s multiple comparisons test of AUC. Asterisks represent statistically significant differences between two groups: * *p* < 0.05, ** *p* < 0.01, *** *p* < 0.001, **** *p* < 0.0001.

**Figure 2 vaccines-12-00052-f002:**
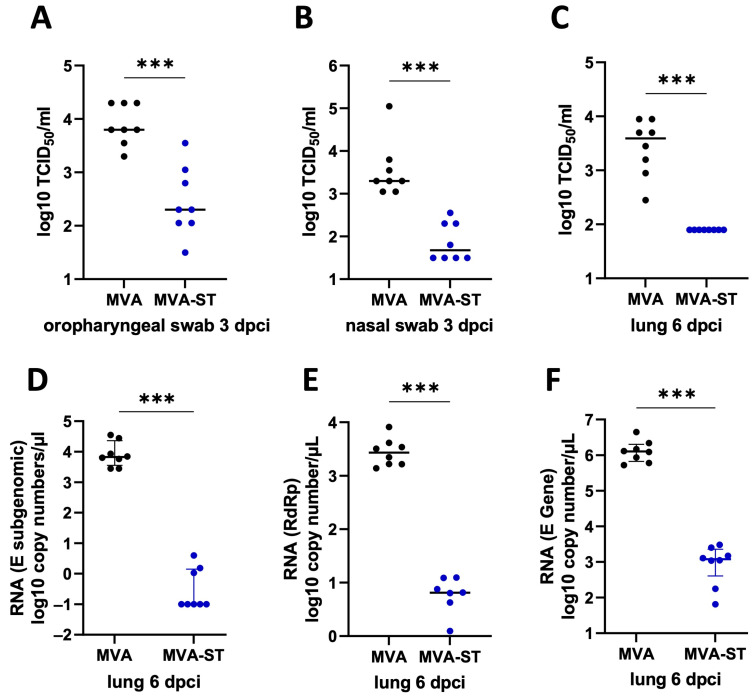
Reduction in viral burden in the lungs induced after MVA-ST immunization in aged Syrian hamsters. **The** MVA-ST-immunized hamsters (*n* = 7) and MVA-control-immunized hamsters (=7) were challenged with 10^4^ TCID_50_ SARS-CoV-2 Bav Pat1 isolate via the intranasal route. Body weight changes, clinical scores, viral loads, pathology and immunogenicity were determined. (**A**) Oropharyngeal and (**B**) nasal swabs were taken on day 3 after challenge infection and evaluated for the titers of SARS-CoV-2 by TCID_50_. (**C**) On the day of death, lungs were prepared and evaluated for the amounts of infectious SARS-CoV-2 using TCID_50_/gram lung tissue and (**D**–**F**) using different PCR-assays for SARS-CoV-2 gRNA copies. Differences between groups were analyzed by Dunn’s multiple-comparisons test. Asterisks represent statistically significant differences between two groups: *** *p* < 0.001.

**Figure 3 vaccines-12-00052-f003:**
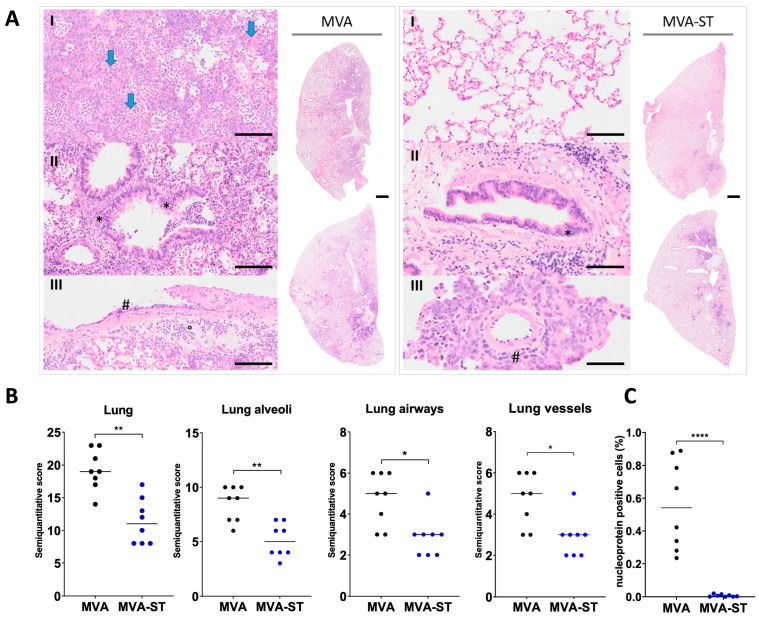
Histopathological characterization of lungs from MVA-ST- and MVA-control vaccinated animals after SARS-CoV-2 challenge 6 dpi. (**A**) Representative overview images of hematoxylin-and-esosin-stained lung sections at 6 dpi, including high-magnification images of alveoli, airways and vessels from MVA-vaccinated and MVA-ST-vaccinated animals. (**I**) Alveolar lesions of lungs from MVA-vaccinated hamsters were characterized by luminal and septal infiltration, mostly of heterophils, macrophages and hemorrhages, which obscured the alveolar architecture (arrows). In alveoli from MVA-ST-vaccinated hamsters, the histopathological lesions were significantly reduced or absent. (**II**) For MVA-vaccinated hamsters, conductive airways frequently showed epithelial hyperplasia (*), and the lumen was often filled with degenerated heterophils, sloughed epithelial cells and macrophages (<). There was moderate peribronchial mononuclear infiltration. Airway histopathological lesions were significantly reduced or absent in the MVA-ST-vaccinated hamsters (**III**) Vascular lesions mainly consisted of histiocytic-heterophilic perivascular and intramural infiltrates (#). There was abundant perivascular edema and perivascular cuffing (°). Vascular histopathological lesions were significantly reduced or absent in the MVA-ST-vaccinated hamsters. Hematoxylin-and-eosin stain, scale bars: 1 mm (overview pictures), 40 µm (high magnifications). (**B**) Lung lesions were graded using semiquantitative scoring systems for histopathological lesions in lungs. (**C**) Nucleoprotein-positive cells were detected by immunohistostaining. Differences between groups analyzed by *t*-Test. Asterisks represent statistically significant differences between two groups: * *p* < 0.05, ** *p* < 0.01, **** *p* < 0.0001.

**Figure 4 vaccines-12-00052-f004:**
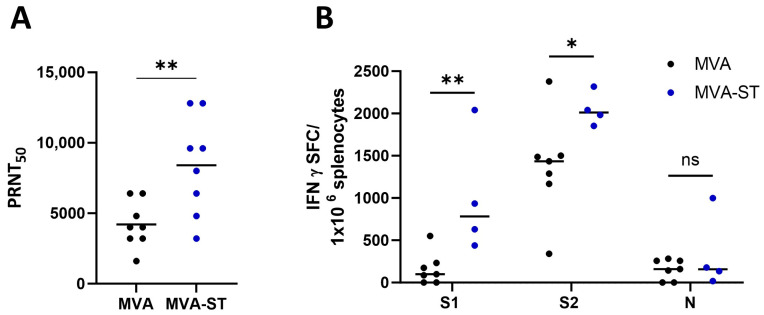
Immune responses after MVA-ST vaccination in serum and spleen. Six days after the intranasal SARS-CoV-2 infection, sera and splenocytes were collected and evaluated for (**A**) SARS-CoV-2-neutralization titers measured by plaque-reduction assay (PRNT_50_) and (**B**) SARS-CoV-2-specific T cells in ELISPOT assays. Differences between groups were analyzed by Mann–Whitney test. Asterisks represent statistically significant differences between two groups: * *p* < 0.05, ** *p* < 0.01.

**Figure 5 vaccines-12-00052-f005:**
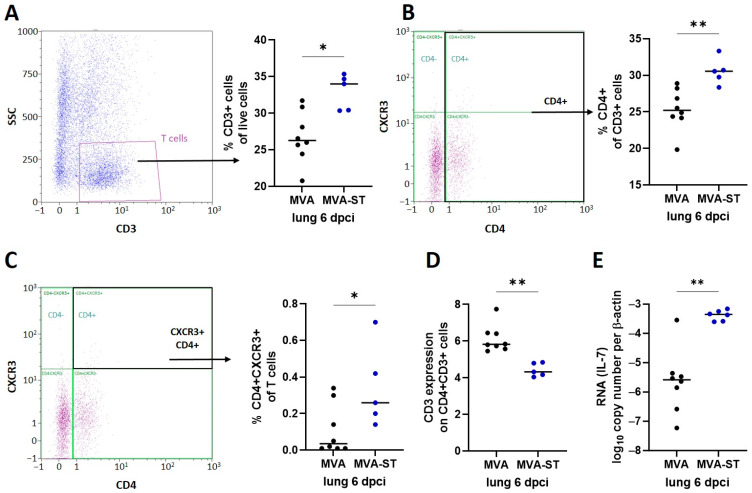
Immune responses in the lung after MVA-ST vaccination. Six days after SARS-CoV-2 infection, lung cells were prepared and evaluated in flow cytometry for (**A**) CD3-, (**B**) CD4- and (**C**) CXCR3-specific antibodies. (**D**) CD3 expression on CD3 + CD4+-positive lung cells shown by median fluorescence intensity (MFI) and (**E**) levels of IL-7 in lungs measured by qRT-PCR. Differences between groups analyzed by t-Test, Mann–Whitney test or two-way ANOVA. Asterisks represent statistically significant differences between two groups: * *p* < 0.05, ** *p* < 0.01.

## Data Availability

The datasets generated during and/or analyzed during the current study are available from the corresponding authors upon reasonable request.

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
