# Peer review of "Protective MVA-ST Vaccination Robustly Activates T Cells and Antibodies in an Aged-Hamster Model for COVID-19"

_vaccines, 2024, doi:10.3390/vaccines12010052_

Round 1
Reviewer 1 Report
Comments and Suggestions for Authors
This study investigated the efficacy of the MVA-ST vaccine in aged Syrian hamsters challenged with SARS-CoV-2. Vaccinated hamsters showed no clinical disease or weight loss and developed significant neutralizing antibodies. Post-infection, control animals exhibited weight loss and clinical symptoms, whereas MVA-ST vaccinated hamsters experienced minimal symptoms and significantly reduced viral titers. Pathological analysis revealed reduced lung damage in vaccinated animals. The study concluded that MVA-ST vaccination effectively protects aged hamsters from COVID-19, inducing robust SARS-CoV-2 specific immune responses.
The study is carefully conducted, and the results sustain the conclusions. The study is of interest considering that age is a major factor regarding the clinical outcome of Covid-19 disease.
My only concern is the comparison with similar studies using Syrian hamsters as an animal model for the testing of MVA-Sars-CoV-2 S recombinants. It would be useful to make at least some comparison between those studies and the present one, or at least mention the corresponding articles, of which at least four are closely related to this manuscript:
Boudewijns R, Pérez P, Lázaro-Frías A, Van Looveren D, Vercruysse T, Thibaut HJ, Weynand B, Coelmont L, Neyts J, Astorgano D, Montenegro D, Puentes E, Rodríguez E, Dallmeier K, Esteban M, García-Arriaza J. MVA-CoV2-S Vaccine Candidate Neutralizes Distinct Variants of Concern and Protects Against SARS-CoV-2 Infection in Hamsters. Front Immunol. 2022 Mar 16;13:845969. doi: 10.3389/fimmu.2022.845969. PMID: 35371064; PMCID: PMC8966703.
Abdelnabi R, Pérez P, Astorgano D, Albericio G, Kerstens W, Thibaut HJ, Coelmont L, Weynand B, Labiod N, Delgado R, Montenegro D, Puentes E, Rodríguez E, Neyts J, Dallmeier K, Esteban M, García-Arriaza J. Optimized vaccine candidate MVA-S(3P) fully protects against SARS-CoV-2 infection in hamsters. Front Immunol. 2023 Oct 18;14:1163159. doi: 10.3389/fimmu.2023.1163159. PMID: 37920464; PMCID: PMC10619667.
Lorenzo MM, Marín-López A, Chiem K, Jimenez-Cabello L, Ullah I, Utrilla-Trigo S, Calvo-Pinilla E, Lorenzo G, Moreno S, Ye C, Park JG, Matía A, Brun A, Sánchez-Puig JM, Nogales A, Mothes W, Uchil PD, Kumar P, Ortego J, Fikrig E, Martinez-Sobrido L, Blasco R. Vaccinia Virus Strain MVA Expressing a Prefusion-Stabilized SARS-CoV-2 Spike Glycoprotein Induces Robust Protection and Prevents Brain Infection in Mouse and Hamster Models. Vaccines (Basel). 2023 May 21;11(5):1006. doi: 10.3390/vaccines11051006. PMID: 37243110; PMCID: PMC10220993.
Wussow F, Kha M, Kim T, Ly M, Yll-Pico M, Kar S, Lewis MG, Chiuppesi F, Diamond DJ. Synthetic multiantigen MVA vaccine COH04S1 and variant-specific derivatives protect Syrian hamsters from SARS-CoV-2 Omicron subvariants. NPJ Vaccines. 2023 Mar 16;8(1):41. doi: 10.1038/s41541-023-00640-y. PMID: 36928589; PMCID: PMC10018591.
Author Response
Point-by-Point response to the reviewers`comments vaccines-2763404
We thank the Reviewers for the excellent critical comments and suggestions. We agree with the Reviewers that these are important questions that would need to be addressed in future hamster studies. We also agree that our study can only provide preliminary results creating the basis for future immunization studies in aged hamsters and other aged animal models. Primary goal of the current study was to address in the aged hamster model whether MVA-SARS-2-ST candidate vaccine can stimulate SARS-CoV-2 specific immune responses and whether such immune responses can provide protection after intranasal SARS-CoV-2 challenge infection. Thus, the current study was initially designed as a proof-of-concept study directly comparing the effects of MVA-ST vaccination and empty MVA-vector control in aged hamsters. Due to time reason we did not succeed in performing additional hamster experiments evaluating the immunogenicity and efficacy of MVA-ST vaccination side-by-side in different age groups of hamsters. This data may help to encourage future MVA-ST-vaccination studies to also include different age groups of hamsters in the same experimental setting. The introduction and discussion section of the revised manuscript indicates now more obviously the major shortcoming of the study that we did not side by side include different age groups of hamsters (young, adult and aged hamsters).
Based on the reviewers comments and to be more clear about the main findings and additionally because of the Reviewer’s suggestion to reflect the limited scope and interpretability of the study, we have made extensive modifications throughout the manuscript in the figures, abstract, introduction and discussion sections. We also changed the title to:” Protective MVA-ST vaccination robustly activates T cells and antibodies in an aged-hamster model for COVID-19”. Throughout the manuscript, we now directly compare the MVA-ST vaccinated aged hamsters with the MVA-control vaccinated aged hamsters. The introduction and discussion section has been essentially re-written. As now stated in the introduction section, we explain the state of the art of vaccination in elderly individuals as well as the current knowledge on the outcome of SARS-CoV-2 infection in elderly individuals. A discussion about the effect of MVA-ST vaccination in aged hamsters has been included in the revised manuscript. We changed the structure of the discussion section to more closely focus on the comparative outcome of MVA-ST and MVA-control vaccination in the aged animals. Moreover, we also extensively addressed the limitations of this study including the lacking side-by-side comparison of the MVA-ST vaccination in different age groups with and without SARS-CoV-2 challenge infections.
We hope that some of these manuscript modifications address the Reviewer’s concerns. All points and limitations addressed by the reviewers have been enabling us to resubmit an improved manuscript. Below we have listed all the reviewers’ points in black and our answers are in blue. Within the manuscript, edits in response to comments are highlighted in yellow.
This study investigated the efficacy of the MVA-ST vaccine in aged Syrian hamsters challenged with SARS-CoV-2. Vaccinated hamsters showed no clinical disease or weight loss and developed significant neutralizing antibodies. Post-infection, control animals exhibited weight loss and clinical symptoms, whereas MVA-ST vaccinated hamsters experienced minimal symptoms and significantly reduced viral titers. Pathological analysis revealed reduced lung damage in vaccinated animals. The study concluded that MVA-ST vaccination effectively protects aged hamsters from COVID-19, inducing robust SARS-CoV-2 specific immune responses.
The study is carefully conducted, and the results sustain the conclusions. The study is of interest considering that age is a major factor regarding the clinical outcome of Covid-19 disease.
My only concern is the comparison with similar studies using Syrian hamsters as an animal model for the testing of MVA-Sars-CoV-2 S recombinants. It would be useful to make at least some comparison between those studies and the present one, or at least mention the corresponding articles, of which at least four are closely related to this manuscript:
Boudewijns R, Pérez P, Lázaro-Frías A, Van Looveren D, Vercruysse T, Thibaut HJ, Weynand B, Coelmont L, Neyts J, Astorgano D, Montenegro D, Puentes E, Rodríguez E, Dallmeier K, Esteban M, García-Arriaza J. MVA-CoV2-S Vaccine Candidate Neutralizes Distinct Variants of Concern and Protects Against SARS-CoV-2 Infection in Hamsters. Front Immunol. 2022 Mar 16;13:845969. doi: 10.3389/fimmu.2022.845969. PMID: 35371064; PMCID: PMC8966703.
Abdelnabi R, Pérez P, Astorgano D, Albericio G, Kerstens W, Thibaut HJ, Coelmont L, Weynand B, Labiod N, Delgado R, Montenegro D, Puentes E, Rodríguez E, Neyts J, Dallmeier K, Esteban M, García-Arriaza J. Optimized vaccine candidate MVA-S(3P) fully protects against SARS-CoV-2 infection in hamsters. Front Immunol. 2023 Oct 18;14:1163159. doi: 10.3389/fimmu.2023.1163159. PMID: 37920464; PMCID: PMC10619667.
Lorenzo MM, Marín-López A, Chiem K, Jimenez-Cabello L, Ullah I, Utrilla-Trigo S, Calvo-Pinilla E, Lorenzo G, Moreno S, Ye C, Park JG, Matía A, Brun A, Sánchez-Puig JM, Nogales A, Mothes W, Uchil PD, Kumar P, Ortego J, Fikrig E, Martinez-Sobrido L, Blasco R. Vaccinia Virus Strain MVA Expressing a Prefusion-Stabilized SARS-CoV-2 Spike Glycoprotein Induces Robust Protection and Prevents Brain Infection in Mouse and Hamster Models. Vaccines (Basel). 2023 May 21;11(5):1006. doi: 10.3390/vaccines11051006. PMID: 37243110; PMCID: PMC10220993.
Wussow F, Kha M, Kim T, Ly M, Yll-Pico M, Kar S, Lewis MG, Chiuppesi F, Diamond DJ. Synthetic multiantigen MVA vaccine COH04S1 and variant-specific derivatives protect Syrian hamsters from SARS-CoV-2 Omicron subvariants. NPJ Vaccines. 2023 Mar 16;8(1):41. doi: 10.1038/s41541-023-00640-y. PMID: 36928589; PMCID: PMC10018591.
We greatly appreciate the reviewer’s positive and constructive feedback. We totally agree with this Reviewer that a comparison with the listed studies and a discussion of these results in the context of our data will significantly improve our manuscript. We included appropriate sections within the introduction and discussion section as follows:
Introduction:
lines 55-59: In recent studies, MVA has been also established as viral vector vaccine against SARS-CoV-2. In that context, all MVA-based COVID-19 candidate vaccines used the SARS-CoV-2 spike protein in a prefusion-stabilized conformation (ST-protein) as vaccine antigen. These different MVA-COVID-19 vaccines confirmed protective efficacy in different preclinical models.
lines 67-69: Based on these previous data, we hypothesize that out MVA-ST candidate vaccine might be also suited as a safe and effective COVID-19 vaccine in an aged hamster model for SARS-CoV-2.
Discussion:
lines 429-434: In agreement with data from previous studies evaluating the effects of MVA-based COVID-19 vaccinations in different preclinical models in different vaccination sched-ules [18-20, 29], we confirmed the activation of SARS-CoV-2 specific immune response and protection in aged MVA-ST-vaccinated hamsters compared to the control aged hamsters which had been vaccinated with empty MVA-vector.
lines 465-472: This is in line with previous studies confirming the robust immunogenicity and efficacy of these MVA-based candidate vaccines against COVID-19. An interesting aspect of these studies is that MVA-ST vaccination also resulted in robust immunogenicity and protection against different SARS-CoV-2 viral variants [18, 20, 29]. Since this is also an extraordinary important aspect of safe and effective COVID-19 vaccines for elderly individuals, it will be important to also evaluate the effects of MVA-ST vaccination in aged hamsters against different SARS-CoV-2 variants.
Reviewer 2 Report
Comments and Suggestions for Authors
The authors used old hamsters as a model for old humans to examine the effect of a vaccinia virus-based MVA vaccine on protection against SARS-CoV-2 infection. The data show that MVA-ST acts in defense against SARS-CoV-2 infection in old hamsters. Both humoral and cellular immunity appear to be involved in this defense.
1. To discuss the usefulness of MVA-ST in older hamsters, it is necessary to compare it with its usefulness in young and adult animals. This paper only presents data on old hamsters, but should be discussed in comparison with data on younger animals.
2. Only a SARS-CoV-2 strain (Germany/BavPat1/2020, NR-52370) was used in this paper, but there are a lot of SARS-CoV-2 variants. The effect of MVA-ST on other variant strains should be discussed.
3. Fig.1C:Because there are no weight data for animals not infected with SARS-CoV-2, it is not clear whether the weight loss of hamsters is influenced by SARS-CoV-2 infection.
4. Fig.3: Please indicate the days post-infection of SARS-CoV-2 (6dpi?). Magnified views of any histopathologic changes that should be emphasized are required.
5. Line335: A clear state for the reason for focusing on IL7 is needed.
Author Response
We thank the Reviewers for the excellent critical comments and suggestions. We agree with the Reviewers that these are important questions that would need to be addressed in future hamster studies. We also agree that our study can only provide preliminary results creating the basis for future immunization studies in aged hamsters and other aged animal models. Primary goal of the current study was to address in the aged hamster model whether MVA-SARS-2-ST candidate vaccine can stimulate SARS-CoV-2 specific immune responses and whether such immune responses can provide protection after intranasal SARS-CoV-2 challenge infection. Thus, the current study was initially designed as a proof-of-concept study directly comparing the effects of MVA-ST vaccination and empty MVA-vector control in aged hamsters. Due to time reason we did not succeed in performing additional hamster experiments evaluating the immunogenicity and efficacy of MVA-ST vaccination side-by-side in different age groups of hamsters. This data may help to encourage future MVA-ST-vaccination studies to also include different age groups of hamsters in the same experimental setting. The introduction and discussion section of the revised manuscript indicates now more obviously the major shortcoming of the study that we did not side by side include different age groups of hamsters (young, adult and aged hamsters).
Based on the reviewers comments and to be more clear about the main findings and additionally because of the Reviewer’s suggestion to reflect the limited scope and interpretability of the study, we have made extensive modifications throughout the manuscript in the figures, abstract, introduction and discussion sections. We also changed the title to:” Protective MVA-ST vaccination robustly activates T cells and antibodies in an aged-hamster model for COVID-19”. Throughout the manuscript, we now directly compare the MVA-ST vaccinated aged hamsters with the MVA-control vaccinated aged hamsters. The introduction and discussion section has been essentially re-written. As now stated in the introduction section, we explain the state of the art of vaccination in elderly individuals as well as the current knowledge on the outcome of SARS-CoV-2 infection in elderly individuals. A discussion about the effect of MVA-ST vaccination in aged hamsters has been included in the revised manuscript. We changed the structure of the discussion section to more closely focus on the comparative outcome of MVA-ST and MVA-control vaccination in the aged animals. Moreover, we also extensively addressed the limitations of this study including the lacking side-by-side comparison of the MVA-ST vaccination in different age groups with and without SARS-CoV-2 challenge infections.
We hope that some of these manuscript modifications address the Reviewer’s concerns. All points and limitations addressed by the reviewers have been enabling us to resubmit an improved manuscript. Below we have listed all the reviewers’ points in black and our answers are in blue. Within the manuscript, edits in response to comments are highlighted in yellow.
The authors used old hamsters as a model for old humans to examine the effect of a vaccinia virus-based MVA vaccine on protection against SARS-CoV-2 infection. The data show that MVA-ST acts in defense against SARS-CoV-2 infection in old hamsters. Both humoral and cellular immunity appear to be involved in this defense.
We thank the Reviewer for the excellent critical comments and suggestions. We agree with the Reviewer that these are important questions that would need to be addressed. We also agree with the Reviewer that the lacking analysis of “normal-aged” hamsters is the main caveats of the study. As indicated above, we have made extensive modifications throughout the manuscript, including revised wording to make the proof-of-concept character of the study clearer and to highlight the goals of this study. Major and minor changes are shown in yellow in the marked-up version of the revised manuscript. The discussion section has been essentially re-written to be clearer in the limitations and the main conclusions.
To discuss the usefulness of MVA-ST in older hamsters, it is necessary to compare it with its usefulness in young and adult animals. This paper only presents data on old hamsters, but should be discussed in comparison with data on younger animals.
We agree with the Reviewer that considering a direct comparison of MVA-ST vaccination in different age groups is necessary to finally discuss its usefulness. To make this more clear, we changed the wording and the structure of the discussion appropriately and included a section within the discussion to highlight the limitations. In doing so, we also changed the title to ” Protective MVA-ST vaccination robustly activates T cells and antibodies in an aged-hamster model for COVID-19”. In the revised version of the manuscript, we focus on our study design, which only included a direct comparison of the safety, immunogenicity and efficacy of MVA-ST vaccination in aged hamsters. Based on this point, we changed the overall structure of the discussion to reflect our study design. We hypothesize that future studies directly comparing the effect of immunizations in different aged animals will be important to contribute to overcome immunosenescence. We highlighted it in the discussion section.
- Only a SARS-CoV-2 strain (Germany/BavPat1/2020, NR-52370) was used in this paper, but there are a lot of SARS-CoV-2 variants. The effect of MVA-ST on other variant strains should be discussed.
This was an excellent suggestion. We agree with the Reviewer that it is important to also evaluate the effect against other SARS-CoV-2 variants and discussed this point based on previous data. Based on the Reviewers comment we have included a section in the discussion on the effect of MVA-ST vaccination against other SARS-CoV-2 variants.
lines 465-472: This is in line with previous studies confirming the robust immunogenicity and efficacy of these MVA-based candidate vaccines against COVID-19. An interesting aspect of these studies is that MVA-ST vaccination also resulted in robust immunogenicity and protection against different SARS-CoV-2 viral variants [18, 20, 29]. Since this is also an extraordinary important aspect of safe and effective COVID-19 vaccines for elderly individuals, it will be important to also evaluate the effects of MVA-ST vaccination in aged hamsters against different SARS-CoV-2 variants.
3. Fig.1C:Because there are no weight data for animals not infected with SARS-CoV-2, it is not clear whether the weight loss of hamsters is influenced by SARS-CoV-2 infection.
We agree with the Reviewer that hamsters not infected with SARS-CoV-2 would have been the best control to confirm the effects of SARS-CoV-2 infection in the aged hamsters. However, we monitored the aged hamsters already before and after the vaccination for a total of 57 days without any SARS-CoV-2 challenge infection and did not observe any effect on weight loss or clinical outcome in these animals. From these data, we assume that the effect on weight loss and clinical disease outcome as observed after SARS-CoV-2 challenge infection is specifically induced by SARS-CoV-2 infection. We included the data from body weight and clinical symptoms in Supplementary Figure 1. To make this conclusion more clear, we also included the following sentence in the discussion section:
lines 452-455: The clinical disease outcome as measured in the control-vaccinated hamsters and also to a minimal content in the MVA-ST vaccinated hamsters can be correlated to SARS-CoV-2 intranasal challenge infection, since we did not detect any clinical disease outcome in the observation period before.
lines 458-460: Again, the outcome of lung pathology is induced by SARS-CoV-2 infections, since other studies did not observe relevant pathology going on in the lung of aged hamsters as background lesion.
4. Fig.3: Please indicate the days post-infection of SARS-CoV-2 (6dpi?). Magnified views of any histopathologic changes that should be emphasized are required.
This was a good suggestion. Based on the Reviewers comment we modified Figure 3A and included the 6 dpi as the time point of necropsy. Moreover, we also included magnified views of the important areas (Lung alveoli, Lung airways, Lung vessels) within the lungs to emphasize the results. We modified the figure legend appropriately and included another section within the results:
lines 327-336: In the lungs of MVA-vaccinated hamsters, we detected epithelial hyperplasia in the conductive airways and moderate numbers of macrophages, degenerated heterophils and sloughed epithelial cells within the airway lumen. Similar lesions in the conduc-tive airways were significantly reduced or absent in MVA-ST vaccinated hamsters (Fig.3A-B). In the lungs of MVA-control vaccinated hamsters, we detected significantly marked vascular lesions compared to the MVA-ST vaccinated hamsters in which they were almost absent. Vascular lesions were mainly consisting of histiocytic-heterophilic perivascular and intramural infiltrates including abundant perivascular edema and perivascular cuffing (Fig.3A-B).
5. Line335: A clear state for the reason for focusing on IL7 is needed.
This was a good suggestion. Based on the Reviewers comment we included a section within the discussion more closely explaining the effect and role of IL7 for the outcome of immunosenescence.
lines 497-506: To support this, MVA-ST-vaccinated hamsters mounted significantly increased levels of Interleukin-7 (IL-7). Diminished activation of proinflammatory cytokines and especially IL-7 have been confirmed to be involved in the outcome of immunosenescence [39]. Here, IL-7 is important and critically involved in the homeostasis of the immune sys-tem [40]. Reduced levels of IL-7 have been correlated to the decline and dysfunction of the immune system [41]. The other way round, robust activation of IL-7 signaling positively affects the maturation and activation of adaptive immune responses including B cells and T cells [42, 43]. In line with this, advanced activation of the IL-7 might explain the improved activation of antigen-specific T cells in MVA-ST-vaccinated animals.
Reviewer 3 Report
Comments and Suggestions for Authors
Clever and colleagues employ an aged Syrian hamster model to show safety and efficacy of a modified vaccinia virus Ankara (MVA) vaccine for SARS-CoV-2. This study is an elegant tour de force with strong relevance and data organization. The methodology is sound and clearly written. The virus-specific MVA-ST vaccine drives a robust immune response that is protective against challenge relative to the vehicle control in aged Syrian hamsters. However, the key premise that this vaccine that can overcome immunosenescence is based on comparison to the group’s previously published data (Ref. #10) without direct statistical analysis (normal adult vs. aged hamsters) to undergird such claims or minimize confounding variables.
STUDY DESIGN CONCERN:
· Ref. 4 (PMID: 35714615) shows a clear difference in immunity of young and aged hamsters, specifically a reduction in the potency of neutralizing antibodies in aged animals, following infection of immunologically naive animals with SARS-CoV-2.
· The authors state in lines 353-356: “we observed altered protection in aged MVA-ST-vaccinated hamsters compared to the equivalent adult hamster model [10]. Our results demonstrated that the immunogenicity and protective outcome of MVAT-ST-immunization in the old hamsters was different from the MVA-ST immunization in normal aged hamsters.” This claim is not apparent based on comparison of data shown in Ref. 10 and the current manuscript—possibly due to different scales on the Y axes). Both manuscripts utilize the same experimental design (prime-boost immunization and challenge dose/route/time). It is this reviewer’s opinion that the immunogenicity and challenge data comparing the normal adult and aged models are equivocal among the MVA-ST-immunized groups. Direct comparisons of ‘adult’ and ‘aged’ hamsters are needed from the same experiment with appropriate statistical analysis for this claim to be scientifically sound. Moreover, the following statement from lines 357-358 appears to contradict the previous statement (lines 353-356, quoted above): “Here, we confirmed significant immunogenicity and efficacy of MVA-ST-prime-boost immunizations [10].”
· To resolve this issue, the authors should consider conducting additional experiments to directly substantiate the impact of age on immunogenicity (prime-boost antibody titers & post-challenge splenocyte reactivity) and viral pathogenesis (swab titers and lung histopathology) among MVA-ST-immunized hamsters. Inclusion of this important control as a supplemental figure would enable direct head-to-head comparisons of normal adult and aged animals and improve the quality of this manuscript.
MINOR COMMENTS:
1. MVA should be defined in the abstract.
2. The humility is appreciated, but it would be appropriate to add the corresponding author’s well-cited review on MVA vaccines (PMID: 28057259) to the introduction as a general reference on the topic.
·3. The authors should comment on the relative degree of histopathology/infiltrates in the lungs (see figure 3) of aged non-challenged animals. Is the semi-quantitative score relative to tissue from healthy age-matched controls or younger animals?
4. How was relative CD3 expression normalized for Figure 5D? Should the MFI (median fluorescence intensity) be reported rather than an apparently arbitrary number?
Author Response
We thank the Reviewers for the excellent critical comments and suggestions. We agree with the Reviewers that these are important questions that would need to be addressed in future hamster studies. We also agree that our study can only provide preliminary results creating the basis for future immunization studies in aged hamsters and other aged animal models. Primary goal of the current study was to address in the aged hamster model whether MVA-SARS-2-ST candidate vaccine can stimulate SARS-CoV-2 specific immune responses and whether such immune responses can provide protection after intranasal SARS-CoV-2 challenge infection. Thus, the current study was initially designed as a proof-of-concept study directly comparing the effects of MVA-ST vaccination and empty MVA-vector control in aged hamsters. Due to time reason we did not succeed in performing additional hamster experiments evaluating the immunogenicity and efficacy of MVA-ST vaccination side-by-side in different age groups of hamsters. This data may help to encourage future MVA-ST-vaccination studies to also include different age groups of hamsters in the same experimental setting. The introduction and discussion section of the revised manuscript indicates now more obviously the major shortcoming of the study that we did not side by side include different age groups of hamsters (young, adult and aged hamsters).
Based on the reviewers comments and to be more clear about the main findings and additionally because of the Reviewer’s suggestion to reflect the limited scope and interpretability of the study, we have made extensive modifications throughout the manuscript in the figures, abstract, introduction and discussion sections. We also changed the title to:” Protective MVA-ST vaccination robustly activates T cells and antibodies in an aged-hamster model for COVID-19”. Throughout the manuscript, we now directly compare the MVA-ST vaccinated aged hamsters with the MVA-control vaccinated aged hamsters. The introduction and discussion section has been essentially re-written. As now stated in the introduction section, we explain the state of the art of vaccination in elderly individuals as well as the current knowledge on the outcome of SARS-CoV-2 infection in elderly individuals. A discussion about the effect of MVA-ST vaccination in aged hamsters has been included in the revised manuscript. We changed the structure of the discussion section to more closely focus on the comparative outcome of MVA-ST and MVA-control vaccination in the aged animals. Moreover, we also extensively addressed the limitations of this study including the lacking side-by-side comparison of the MVA-ST vaccination in different age groups with and without SARS-CoV-2 challenge infections.
We hope that some of these manuscript modifications address the Reviewer’s concerns. All points and limitations addressed by the reviewers have been enabling us to resubmit an improved manuscript. Below we have listed all the reviewers’ points in black and our answers are in blue. Within the manuscript, edits in response to comments are highlighted in yellow.
Clever and colleagues employ an aged Syrian hamster model to show safety and efficacy of a modified vaccinia virus Ankara (MVA) vaccine for SARS-CoV-2. This study is an elegant tour de force with strong relevance and data organization. The methodology is sound and clearly written. The virus-specific MVA-ST vaccine drives a robust immune response that is protective against challenge relative to the vehicle control in aged Syrian hamsters. However, the key premise that this vaccine that can overcome immunosenescence is based on comparison to the group’s previously published data (Ref. #10) without direct statistical analysis (normal adult vs. aged hamsters) to undergird such claims or minimize confounding variables.
STUDY DESIGN CONCERN:
- Ref. 4 (PMID: 35714615) shows a clear difference in immunity of young and aged hamsters, specifically a reduction in the potency of neutralizing antibodies in aged animals, following infection of immunologically naive animals with SARS-CoV-2.
- The authors state in lines 353-356: “we observed altered protectionin aged MVA-ST-vaccinated hamsters compared to the equivalent adult hamster model [10]. Our results demonstrated that the immunogenicity and protective outcome of MVAT-ST-immunization in the old hamsters was different from the MVA-ST immunization in normal aged hamsters.” This claim is not apparent based on comparison of data shown in Ref. 10 and the current manuscript—possibly due to different scales on the Y axes). Both manuscripts utilize the same experimental design (prime-boost immunization and challenge dose/route/time). It is this reviewer’s opinion that the immunogenicity and challenge data comparing the normal adult and aged models are equivocal among the MVA-ST-immunized groups. Direct comparisons of ‘adult’ and ‘aged’ hamsters are needed from the same experiment with appropriate statistical analysis for this claim to be scientifically sound. Moreover, the following statement from lines 357-358 appears to contradict the previous statement (lines 353-356, quoted above): “Here, we confirmed significant immunogenicity and efficacy of MVA-ST-prime-boost immunizations [10].”
- To resolve this issue, the authors should consider conducting additional experiments to directly substantiate the impact of age on immunogenicity (prime-boost antibody titers & post-challenge splenocyte reactivity) and viral pathogenesis (swab titers and lung histopathology) among MVA-ST-immunized hamsters. Inclusion of this important control as a supplemental figure would enable direct head-to-head comparisons of normal adult and aged animals and improve the quality of this manuscript.
We greatly appreciate the Reviewer’s critical and constructive feedback. We agree with this Reviewer that the appropriate experimental study design would include a direct comparison of adult and aged hamsters in the same experiment. Since we did not succeed in performing another animal experiment due to time reasons, we tried to revise our initial statements and our wording to appropriately reflecting the results from our study design. Since we only compared the safety, immunogenicity and efficacy of MVA-ST vaccination in aged hamsters without another MVA-ST vaccination group of adult hamsters, we now focused on these results. In doing so, we also changed the title to “Protective MVA-ST vaccination robustly activates T cells and antibodies in an aged-hamster model for COVID-19”. In the revised version of our manuscript, we changed results and directly compare now the aged hamsters in each case for the effect of vaccination on antibodies and T cells and protection. In the revised version of the manuscript, we did not directly compare these data with the results obtained in our previous paper in which we used adult hamsters. Based on this point, we changed the overall structure and focus of the discussion to reflect our study design. We hypothesize that future studies directly comparing the effect of immunizations in different aged animals will be important to contribute to overcome immunosenescence. We highlighted it in the discussion section.
MINOR COMMENTS:
- MVA should be defined in the abstract.
This was a good suggestion. Based on the Reviewers comment we included the definition of MVA into the abstract as follows:
lines 15-17: Modified Vaccinia virus Ankara (MVA) is a highly attenuated and replication deficient Vaccinia virus that has been established as multipurpose viral vector for vaccine development against various infections.
- The humility is appreciated, but it would be appropriate to add the corresponding author’s well-cited review on MVA vaccines (PMID: 28057259) to the introduction as a general reference on the topic.
We highly appreciate the great suggestion and included our review on MVA within the introduction section.
- 3. The authors should comment on the relative degree of histopathology/infiltrates in the lungs (see figure 3) of aged non-challenged animals. Is the semi-quantitative score relative to tissue from healthy age-matched controls or younger animals?
We thank the reviewer for the excellent suggestion. We are aware that this study does not have an age matched-non infected group. However, based on the authors experience in pathological evaluation of SARS-CoV-2 infected hamsters and further supported by the literature (https://journals.sagepub.com/doi/pdf/10.1177/0192623314532569 ) we assume that aged hamsters do not have relevant pathology going on in the lung as background lesion. If there would be a consistent background pathology in the lung of aged hamsters due to age related changes, we hypothesize that both MVA-vaccinated and MVA-ST-vaccinated hamsters would express such changes in the lungs and so it would not bias the experimental outcome. But based on our results from pathological evaluations, this seems not to be the case. The semiquantitative score is a thorough and detailed histopathological score that was applied by veterinary pathologist and that has been already published in multiple works (most of them cited in the reference list) to evaluate lung histopathological changes induced by SARS-CoV-2. The score was made for scoring changes in the lung of multiple animal model of COVID-19 of different stages of life. Our score is detecting changes that are specific for SARS-CoV-2 infection and seems not to be influenced by age. It is based on significant experience in scoring lungs from SARS-CoV-2 infected animals from different animal models and experimental settings, and is well discussed in the supplementary material of another study from the authors: https://www.nature.com/articles/s41467-022-31200-y. Moreover, the authors have already applied this score on aged hamsters (about 1 year old) mock infected with PBS or infected with SARS-CoV-2 in another study that is at the moment in the writing process now. The results from this study further support that there is no impact or influence of the age of the animals. The inflammation in the lung is minimal at 6 dpi in the mock animals (data not shown, manuscript in writing process). Aged hamsters tend to rather have background lesions in other organs such as liver and heart, not much in the lung (data not shown, manuscript in writing process).
- How was relative CD3 expression normalized for Figure 5D? Should the MFI (median fluorescence intensity) be reported rather than an apparently arbitrary number?
This was a good suggestion. Based on the Reviewers comment we modified figure 5D by reporting the MFI and also included it in the Material and Methods section (see lines 237-238).
Round 2
Reviewer 2 Report
Comments and Suggestions for Authors
The authors adequately addressed the reviewer's comments.
Reviewer 3 Report
Comments and Suggestions for Authors
The authors have addressed all of my concerns. One minor editorial point is that the authors modified the manuscript text and legend for figure 5D, the original graph was not replaced with one that directly shows MFI for CD3 in the image.